# Impact of *Prosopis velutina* Wooton on the Composition and Diversity of Native Woody Species in a Semi-Arid Zone along the Molopo River, South Africa

**DOI:** 10.3390/plants12071561

**Published:** 2023-04-05

**Authors:** Makuété A. P. Tiawoun, Pieter W. Malan, Alvino A. Comole, Moleseng C. Moshobane

**Affiliations:** 1Unit for Environmental Sciences and Management, Department of Botany, North-West University, Mafikeng, Private Bag X 2046, Mmabatho 2735, South Africa; pieter.malan@nwu.ac.za (P.W.M.); aa.comole@nwu.ac.za (A.A.C.); 2School of Mathematics, Sciences and Technology Education, Department of Natural Sciences, North-West University, Mafikeng, Private Bag X 2046, Mmabatho 2735, South Africa; 3South African National Biodiversity Institute, Pretoria National Botanical Garden, 2 Cussonia Avenue, Brummeria, Silverton 0184, South Africa; m.moshobane@sanbi.org.za; 4Centre for Functional Biodiversity, School of Life Sciences, University of KwaZulu-Natal, P/Bag X01, Scottsville, Pietermaritzburg 3209, South Africa

**Keywords:** *Prosopis velutina*, invasion status, species composition, species diversity, riparian zone, Molopo River

## Abstract

Invasive alien species represent one of the main threats to biodiversity and species extinction. This is the case for the genus *Prosopis,* among which *Prosopis velutina* is the most invasive and common tree species along the Molopo River in the North-West Province, impacting native plant communities. However, its impact on the composition and diversity of native woody species remains poorly investigated in the area. Thus, this study aimed to assess the impact of *P. velutina* on native woody plant composition and diversity across three sites along the Molopo River. At each site, five quadrats of 20 × 20 m^2^ were randomly established in invaded and adjacent uninvaded stands. A comparative methodological approach was adopted, and the woody plants in invaded and uninvaded stands with similar site conditions were sampled. The results showed that native woody species density differed significantly (*p* < 0.05) between invaded and uninvaded stands, except for Bray sites, where there was a marginal difference (*p* = 0.6). The overall native woody species density decreased by 79.7% in the invaded stand. However, non-metric multidimensional scaling (nMDS) and analysis of similarity (ANOSIM) indicated significant differences in native tree composition between invaded and uninvaded stands at all sites. In all three sites, all ecological indices had significantly lower values in invaded stands compared to uninvaded stands. The decrease in all ecological indices in invaded over uninvaded stands indicated that *P. velutina* invasion reduced the diversity of native woody plant species. Due to the incessant spread of *P. velutina*, it may become a long-term dominant species with an increasing impact on the native vegetation. Therefore, the findings of this study call for urgent management and appropriate control measures against the ongoing spread of this invader within the riparian zones of the Molopo River in North-West Province.

## 1. Introduction

Invasive alien species (IAS) are a devastating threat to global biodiversity, causing the decline or even extinction of native species. The invasion of alien plant species into indigenous plant communities has become a worldwide phenomenon over the past few decades [1] and is rapidly increasing in numbers and distributions [2], to the extent that native species strive to recover. This is known as an ecological problem that negatively affects the conservation of biodiversity. The spread of invasive plant species causes extensive effects on the habitats they invade by altering soil stability, promoting erosion, and inducing various environmental effects that change native plant composition and structure [3,4]. Furthermore, IAS have pronounced negative impacts on human livelihoods by decreasing ecosystem services [5].

Riparian plant communities are biologically diverse and are threatened worldwide [6]. These threats are caused by high rates of natural disturbances, such as water movement and flooding, which cause high nutrient deposition [7,8]. Riparian zones are highly modified in most parts of the world [9], and in some parts of South Africa, invasive alien plants have severely degraded riparian habitats [10]. Alien plant invasions within the riverine system of the Molopo River may have detrimental effects, making these invaded riparian areas unproductive for crops and livestock that require this habitat type for foraging. Riparian zones are highly dynamic systems that are naturally disturbed [11] and are recognized as natural habitat types that are particularly vulnerable to being invaded by exotic plants [11,12].

*Prosopis* species, commonly known as mesquite, from North and South America, were introduced to South Africa and distributed in various areas of the country in the late 1800s for several functions, such as providing shade for livestock, fuel wood, pods for fodder, and wood for construction and furniture production [13,14,15]. *Prosopis* species have since been identified as a serious problem in parts of South Africa and ranked as the second-most widespread invasive plant taxon after Australian *Acacia* species [16]. *Prosopis* species are recognised as highly invasive plants in both their native and introduced ranges [17]. There is evidence that *Prosopis* species are spreading at an alarming rate, and they have since been identified as a serious problem in the country with undesirable ecological and socio-economic consequences. Although they form extensive impenetrable thickets over large areas in the Northern Cape, Western Cape, Free State, and North-West provinces [14,15], they predominantly occupy riparian zones with dense populations. With the ongoing and extensive *Prosopis* invasion in these provinces, it was necessary to locally assess the diversity and composition of the coexisting native woody species alongside the riparian zone of the Molopo River. Hence, the current study assessed the impact of *Prosopis velutina* on the composition and diversity of native woody plant species at three sites along the Molopo River.

Many studies have been conducted to quantify the ecological and socio-economic impacts of alien plant invasions and to develop efficient management approaches [18,19,20]. There is limited knowledge of alien plant impact and occurrence in South Africa [21,22,23]. However, the potential expansion of the invasive plants in the North-West Province remains underexplored, especially within the riparian areas. Despite evidence suggesting that riparian zones are among the natural habitats more prone to the establishment of invasive alien plants [9], the need to protect the valuable and vulnerable resources in these areas is important for conservation of biodiversity [10]. However, very little attention has been paid to understanding the impacts of exotic species on indigenous woody plant communities in riparian zones. In this context, one significant invader plant commonly occurring within the riparian areas alongside the Molopo River in the North-West Province and provoking serious economic and ecologic concerns is *P. velutina* [24,25]. The dominance of *P. velutina* reduces the physiognomic heterogeneity of riparian habitats. Due to the limitlessness of the invaded area and its inaccessibility, no adequate recent information exists about the impact of *P. velutina* on the species composition and diversity of invaded plant communities along the Molopo River. However, in terms of the Alien and Invasive Species Regulations (AIS), National Environmental Management: Biodiversity Act (Act No 10 of 2004), *P. velutina* is listed as a category 1b species in the Eastern and Western Cape, Free State, and North-West Provinces and in the Northern Cape Province as a category 3 status—except for riparian areas, where it is regarded as a category 1b species [26,27].

Thus, comparative studies that may provide crucial information were used to evaluate the impact of *P. velutina* invasions on native plant communities. Therefore, this study aimed at assessing the impacts of this invader on woody species composition and diversity by answering the following questions: (1) Does the invasion of *P. velutina* alter the composition and diversity of native woody plant species? (2) Are the impacts of the *P. velutina* invasion similar across the three sites?

## 2. Results

### 2.1. Species Composition and Density

Across all the three selected sites along the Molopo River, a count of native woody plant species in uninvaded stands (Un) was seven from four families, compared with five woody species from three families in invaded stands (In). Five species from three families were common to the two stands (Table 1). Overall, five species, namely *Senegalia mellifera*, *Vachellia hebeclada*, *V. erioloba*, *Ziziphus mucronate*, and *Melia azedarach*, were found to coexist with *Prosopis velutina*. Among the four families representing all the species, Fabaceae was found to be the most species-rich family in both invaded and uninvaded stands of the entire study area, with four species (Table 1). The invasion of *P. velutina* reduced the number of woody species in the invaded communities by 28.6%, from seven woody species in the uninvaded stand to five woody species in the invaded stand (Table 1). *Vachellia erioloba* was the most abundant of the seven native woody plant species recorded in uninvaded stands of all the three sites and accounted for 100%, 33.1%, and 26.5% of total densities of native woody plants in Bray, Tshidilamolomo, and Mabule, respectively. At the Tshidilamolomo and Mabule sites, the invaded stands were dominated by *S. mellifera* and *V. hebeclada* when *P. velutina* was excluded from the data (Table 1). In each site, invaded stands had a significantly lower woody plant density (TE ha^−1^) compared to uninvaded ones (Figure 1). When compared to uninvaded stands, the density of native woody species decreased by 79.7% in *P. velutina* invaded stands. The invaded and uninvaded stands differed significantly (*p* < 0.05) in density. Invaded stands differed amongst themselves (*p* < 0.05) and the uninvaded stands as well (Table 1).

The combining density for all native woody species in each site was lower in invaded than uninvaded stands by about 92.3% (F = 5.2, *p* = 0.03), 74.3% (F = 2.9, *p* = 0.04), and 42.4% (F = 0.5, *p* = 0.6) in Tshidilamolomo, Mabule, and Bray, respectively. Native woody plant density differed significantly between uninvaded and invaded stands, except at the Bray site where there was a marginal difference (*p* = 0.6). However, a significant difference was found between the three sites (Table 1). The Mabule site had a significantly greater species density than the other two sites (Figure 1). 

At Tshidilamolomo, Mabule, and Bray sites, the ordination nonmetric multidimensional scaling (nMDS) and analysis of similarities (ANOSIM) revealed significant differences in the species composition of invaded and uninvaded stands, with global R values of (R = 1, *p* = 0.0078), (R = 1, *p* = 0.0082), and (R = 0.348, *p* = 0.0423), respectively (Figure 2). A SIMPER analysis of the data revealed that species composition contributed the most to the average dissimilarity between uninvaded and invaded stands. This analysis also computed the average contribution of species causing dissimilarity. SIMPER analysis showed 78.2% overall dissimilarity among invaded and uninvaded stands (Table 2). The top contributing woody species causing differences between uninvaded and invaded stands include *V. erioloba* and *S. mellifera* (Table 2).

### 2.2. Ecological Indices

Since the invasion of *Prosopis velutina* was the main factor distinguishing between invaded and uninvaded stands, this species was excluded from the data of ecological indices (Table 2). The analysis showed that all the ecological indices, such as species richness (R), Shannon diversity (H’), Simpson diversity (D), and species evenness (J), were significantly different between invaded and uninvaded stands at Tshidilamolomo and Mabule, but not significantly different at Bray sites (Table 3). The invaded stands were associated with fewer native woody species than the uninvaded stands. *Prosopis velutina* showed variable impacts across the three sites by decreasing species numbers per stand by a maximum of 67% at Tshidilamolomo and 50% at Mabule, while an equal number of species were found at the Bray site. Furthermore, at Tshidilamolomo and Mabule, all ecological indices were lower in the invaded stand than the uninvaded stand, with significant differences between the two stands (*p* < 0.05). At the Bray site, the values of the ecological indices were roughly equal between invaded and uninvaded stands. A comparison between the invasion categories at the Bray site showed that species richness (*p* = 0.05), Shannon diversity (*p* = 0.06), Simpson index (*p* = 0.06), and species evenness (*p* = 0.06) were not significantly affected (Table 3). Hence, there was no significant difference (*p* > 0. 05) between the two stands at the Bray site (Table 3).

From the three sites, the Tshidilamolomo site was the most severely affected by the invasion of *P. velutina*. The differences in mean species richness, Shannon diversity, Simpson diversity, and Evenness values were higher at the Tshidilamolomo site and accounted for 69%, 50%, 31.5%, and 43% of the reduction in the invaded stands, respectively. In contrast, the Bray site was the least affected by the invasion of *P. velutina.* In this site, species richness, Shannon diversity, Simpson index, and species evenness were reduced by 5.26%, 7.7%, 7.4%, and 4.2%, respectively, in the *P. velutina* invaded stands (Table 3). Nevertheless, the impact of the *P. velutina* invasion did not vary much between Tshidilamolomo and Mabule but did between these two sites and the Bray site (Table 3).

Overall, the invaded stands at the three sites have apparently caused a decline in woody plant species. However, in Tshidilamolomo and Mabule, a significant difference in diversity indices between invaded and uninvaded stands was observed. In the Bray site, the invaded and uninvaded stands showed minimal differences in diversity indices (Table 3). The combined mean evenness index of all the invaded stands and that of the uninvaded were 0.68 and 0.81, respectively. Thus, the heterogeneity of the invaded stands was reduced by 16%. 

## 3. Discussion

Invasive plants have frequently become detrimental to habitat structure by decreasing biodiversity and displacing native species [28]. However, the consequences of biological invasions are not always predictable and must be assessed considering local factors [29]. The current study compared the impact of *Prosopis velutina* on the composition and diversity of native woody plant species under different invasion categories at three sites, namely Tshidilamolomo, Mabule, and Bray, alongside the Molopo River. The results revealed that *P. velutina* has harmful effects on native woody plant species composition and diversity. Hence, woody composition and diversity differ between the invaded and uninvaded stands. The findings indicated that *P. velutina* invasion decreased native woody plant species in the invaded stands. This supports the findings of Eshete et al. [30], who showed that the presence of *Prosopis juliflora* reduced the abundance of native species in the invaded plots. In line with this study, several other studies have shown that *Prosopis* invasion can strongly influence the composition and diversity of adjacent vegetation [31,32,33].

The ordination (nMDS) and ANOSIM analyses of native woody species composition showed significant variations between invaded and uninvaded stands. These differences can be considered as the result of the change in invaded vegetation composition because the invaded stands represent a reduced subset of uninvaded stands, which makes these stands similar in terms of species composition. Although there were no significant differences in species composition and diversity between invaded and uninvaded stands at the Bray site, in Tshidilamolomo and Mabule, there were significant differences between invasion status. Similar to these findings, other studies have found that differences in invasion categories may be due to other factors such as disturbance rather than the presence of the invasive alien species alone, which reduced the number of woody species [34].

A recent study by Tiawoun et al. [25], indicated that *Prosopis* species are among the most aggressive invasive aliens, spreading in arid and semi-arid ecosystems throughout the country. They form dense thickets and can readily outcompete native plants, having an important negative impact on vulnerable native species [32,35]. Many countries in Sub-Saharan Africa and elsewhere are vulnerable to aggressive alien invasive species like *Prosopis* spp. which are well adapted to arid conditions [36]. At the three sites, *P. velutina* revealed variable impacts by reducing all the values of ecological indices over uninvaded stands. A low value for diversity indices in invaded stands at Tshidilamolomo and Mabule sites indicated that the invaded stand at these sites is dominated by a limited number of species. This could be the result of the high invasion of *P. velutina,* with a few woody species being well adapted to that environment and weaker species competing with it. These findings are consistent with other studies on invasive species, which show that invasive species have a significant negative impact on floral composition and diversity [37,38]. Similar modifications were found in Australia, where fewer species were recorded in invaded areas [17]. However, at the Bray site, the value of ecological indices slightly declined with *P. velutina* invasion, and the analysis proved no significant differences between invaded and uninvaded stands. The variation of plant species over different sites could be attributed to several environmental factors that impose impacts on both temporal and spatial scales [39]. Thus, environmental heterogeneity, regeneration capacity, level of disturbance, and competition might shape and determine the species richness of each site. 

The outcomes of the current study revealed the detrimental impacts of *P. velutina* on woody species composition and diversity at the Tshidilamolomo and Mabule sites. Several studies have found that the invasion of *Prosopis* species has reduced native plant richness and density [32,40,41,42,43]. This suggests that the vigorous and rapid growth of this invasive species, which has reached a large proportional representation, is accountable for the strong impact on native woody plant species along the Molopo River. The allelopathic effects of *Prosopis* also play a role in the competitive exclusion of native species from invaded plots [44]. The ability of *P. velutina* to form homogenous stands seems to be another effective means of reducing the survival of native vegetation. The uninvaded stands of *P. velutina* at the Bray site, on the other hand, had a slightly higher value of ecological indices. Moreover, no significant differences in ecological indices between invaded and uninvaded stands were recorded. These results suggest that *P. velutina* exhibits a minor impact on the native woody species of the invaded stands studied. Therefore, it is possible that *P. velutina* invasion does not necessarily cause detrimental changes in the vegetation that was investigated at this site. A similar trend was found in the study of Kumar and Mathur [45], where communities invaded by *Prosopis juliflora* had significantly higher native species richness and diversity in their plant populations in arid grazing lands.

The successful establishment of *P*. *velutina* along the Molopo River may depend on several ecological factors in the ecosystems it invades. Stromberg et al. [46] revealed that several key environmental variables influence the composition of riparian and wetland vegetation growing in the floodplain of a semi-arid river. Furthermore, the invasion of this species may also depend on its biological characteristics, such as the production of large numbers of seeds that may disperse widely and remain viable in the soil for considerable periods of time [36,47,48]. Its deep and extensive root systems lead to the depletion of groundwater in water-scarce surroundings, which causes native trees to dry out [33]. According to Hejda et al. [37], the effect of invasion on native species is mainly species-specific, and the severity of the impact depends almost exclusively on the identity of the particular invading species. Native species differ in their resistance to invasion; hence, some are excluded from invaded communities more easily than others [49,50]. Native species with a high potential to compete for natural resources with invasive species in the invaded plots have a chance of survival [51]. In this study, *Senegalia mellifera* and *Vachellia hebeclada* seem to be such species. Although *P. velutina* reduced their abundances significantly, they were more associated with *P. velutina* than the other woody plant species. According to Comole et al. [24], these native species are quite a constant element of the *P. velutina* association.

## 4. Materials and Methods

### 4.1. Study Area

The study was conducted in three selected sites, namely Tshidilamolomo (25°49′04.3″ S and 24°41′05.60″ E), Mabule (25°46′23.87″ S and 24°33′14.93″ E), and Bray (25 27′41.42″ S and 23°42′05.05″ E), situated in the Ngaka Modiri Molema Municipal District in the North-West Province (NWP) of South Africa (Figure 3). This district forms part of the south-eastern edge of the Kalahari sand basin [52] and is 1000–1300 m above sea level [53].

The study area falls within the Eastern Kalahari Bushveld (SVK 1, Mahikeng Bushveld Vegetation Type) [52], in semi-arid areas of the Savanna Biome. According to Barnes [55], a semi-arid savanna is characterised by a variety of physiognomic vegetation, typical of Africa’s tropical summer rainfall regions. The area receives mean annual precipitation (MAP) ranging from 360 mm to 520 mm, whereas the mean annual temperature varies between −1.8 °C in June and 35.6 °C in November [53]. The main rainy season lasts from November to March [56]. The daily temperature is often 42°C during the summer months, with the winter months being much colder (−9 °C) [57].

Hutton and Clovelly soils are common. The study area is situated in the Ah, Ai, and Ae land types [53,58]. The geology of the study area mainly represents Aeolian Kalahari sand of Tertiary to recent age in flat to sandy soils [53]. Mucina and Rutherford [53] classified the study area as Molopo Bushveld, a semi-arid Kalahari thorn bush-type savanna. Almost no areas along the Molopo River are completely free of *Prosopis* species.

### 4.2. Experimental Design 

Field surveys were performed within three selected sites (Figure 3) after the rainy season because most of the species sprout during this time. The three selected sites cover a wide range of site conditions and vegetation types in which the invader, *Prosopis velutina* (Figure 4), was dominant. These sites were paired to include one stand with *P. velutina* and another without the invader. The comparative stands were selected according to the following criteria: the invaded stand was required to be dominated by *P. velutina*, while in the adjacent uninvaded stand, *P. velutina* was absent or occurred in very low numbers, which could have little or no induced changes to native woody species composition and diversity [1]. Nevertheless, the presence of *P. velutina* serves as evidence that habitat conditions in uninvaded plots were suitable for invasion in the future. According to Richardson et al. [59], invasive species usually only affect native species if they are dominant. Both the invaded and uninvaded stands had to be as similar in terms of vegetation structure as possible, with no obvious differences in soil type.

At each of the sites, the random sampling technique was used based on the availability and abundance of *Prosopis* in invaded stands. Each selected stand (invaded and uninvaded) within each site had 5 quadrats measuring 20 × 20 (400 m^2^) to compare the potential impacts of *P. velutina* on indigenous woody plant species.

### 4.3. Data Analysis

*Prosopis velutina* invasion effects on species composition and diversity were evaluated at all sampling sites, both within invaded and uninvaded stands. To assess differences in woody plant species composition and diversity among stands, *P. velutina* was excluded from the analysis, as the aim was to evaluate its impact on the remaining native woody plant species.

All native woody plant species were identified, and individuals were counted in all invaded and uninvaded studied quadrats. The numbers were summed up to get the total number of plant species and individuals from each stand at each site. To determine the representation of each species relative to the entire plant community, species composition (%) at each site were calculated. 

In each stand, the species composition (%) and the woody density of each native woody plant species were estimated. 

Species composition = Number of individuals of a species/Number of individuals of all species.

Woody density = Total Number of individuals of a species × factor/Area in hectares (ha).

The woody species densities were determined by converting the total number of individuals of each species encountered in each stand to an equivalent number per hectare (TE ha^−1^). The data were analysed according to the height of plants (1 TE = 1 tree of 1.5 m, thus 2 TE = 1 tree of 3 m etc.). A tree equivalent (TE) is defined as a 1.5 m-high tree and is widely used to express tree densities in woody plant population studies [60].

To explore the response of native woody species composition to *Prosopis velutina* invasion, one-way analysis of similarities (ANOSIM) was used to assess differences in woody species composition between invaded and uninvaded stands at the three sites. To check the similarity index in community composition between stands, the Bray–Curtis similarity matrix was performed, using a permutation test with 999 simulations. The values of the Bray–Curtis similarity index fall between 0 (communities are identical) and 1 (two communities are completely dissimilar). Similarity percentages (SIMPER) analysis was used to assess the percentage contribution of each plant species to the overall similarity between invasion conditions. All these were performed based only on native woody species abundance.

To evaluate the impact of *P. velutina* invasion on woody species diversity, ecological indices including species richness (R), Shannon index of diversity (H′), Simpson index of dominance (D), and species evenness (J’) were calculated and compared for invaded and uninvaded stands in each site.

The ecological indices were calculated using the equation below [61]:

Species richness: R = S − 1/lnN.

Shannon diversity index: H′ = −Σn_i_/N ln (n_i_/N).

Simpson index of dominance: D = Σn_i_ (n _i_− 1)/N (N − 1).

Evenness was calculated as J = H′/InS.

S = the total number of species, N = Total number of individuals, ni = number of individuals of the species.

### 4.4. Statistical Analysis

The field inventory data were recorded in a Microsoft Excel 2019 datasheet. Both invaded and uninvaded stands were the independent variables, while density, species richness, species diversities, species evenness, and Simpson index of dominance were considered as dependent variables. The independent variables were subjected to a one-way analysis of variance (ANOVA) with invasion status. Differences between independent variables for the three sites were individually tested for significance between invaded and uninvaded stands. Tukey’s HSD test was used to test for differences between averages where differences between averages were considered significant if *p* < 0.05. The means of ecological indices were reported with standard errors (mean ± SE). All statistical methods were performed using PAST Software, version 13.0.

## 5. Conclusions

Increasing woody alien invasive species around the world poses a major threat to native plant species. The present study clarified the local impact of *Prosopis velutina* on native woody plant species. The study found that *P. velutina* invasion had a detrimental impact on native woody plant species in the studied sites by reducing the number of species, richness, diversity, and evenness. The decrease in ecological diversity indices in invaded over uninvaded stands is an indication that plant communities may become monospecific with *P. velutina* invasion. Thus, this invasive species is likely to become a dominant species with an increasing detrimental impact on the native vegetation. Therefore, this invader plant will cause not only an ecological problem, but also create challenges for local people, such as depleted grazing areas for livestock, the loss of useful plant species, and a threat to the plant diversity of invaded areas. The findings of this study focus on increasing local awareness of the issue and call for urgent management and appropriate control measures against the ongoing spread of this invader within the riparian zones of the Molopo River in the North-West Province.

## Figures and Tables

**Figure 1 plants-12-01561-f001:**
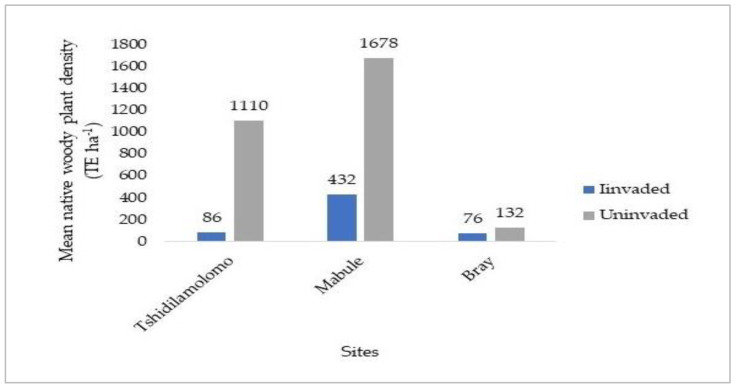
The mean density of native woody plants (TE ha^−1^) in the invaded and uninvaded stands at the three sites along the Molopo River.

**Figure 2 plants-12-01561-f002:**
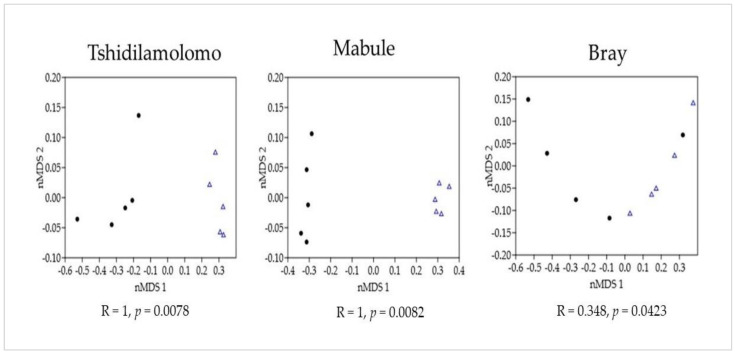
Multidimensional scaling (MDS) ordination and analyses of similarity (ANOSIM) result of invasion status at three sites along the Molopo River. Invaded (•); uninvaded (∆).

**Figure 3 plants-12-01561-f003:**
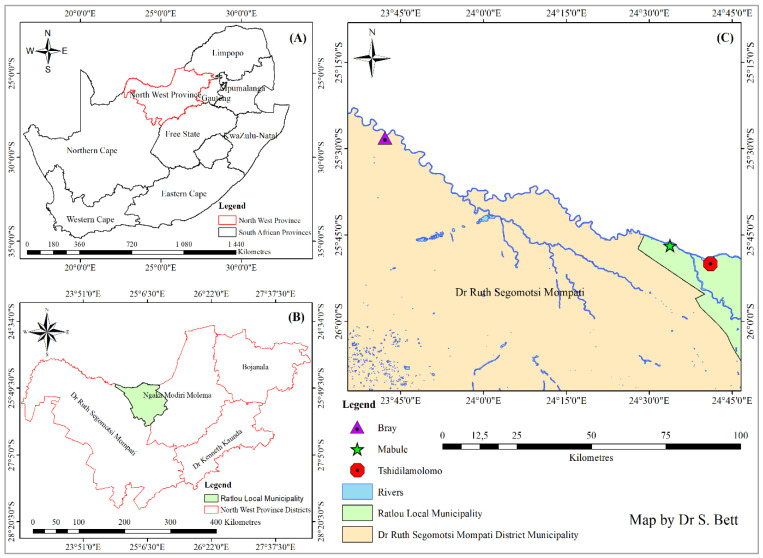
Map showing the three selected study sites along the Molopo River in the Ngaka Modiri Molema District (**C**) in the North-West Province (**B**) of South Africa (**A**) adapted from Tiawoun et al. [54].

**Figure 4 plants-12-01561-f004:**
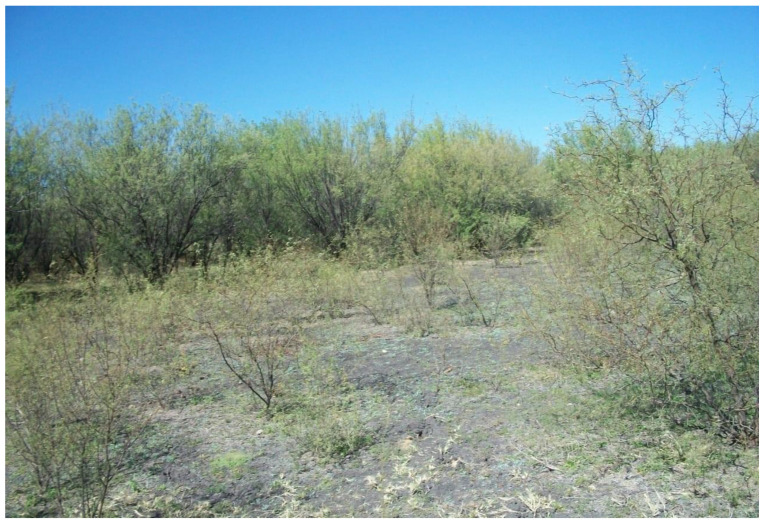
*Prosopis velutina* in one of the selected sites along the riverine system of the Molopo River.

**Table 1 plants-12-01561-t001:** Density and proportional representation of each native woody plant species recorded at three different sites between *Prosopis velutina* invaded (In) and uninvaded (Un) stands.

Woody Species	Family	Tshidilamolomo				Mabule				Bray			
		Density (TE ha^−1^)		Composition (%)		Density (TE ha^−1^)		Composition (%)		Density (TE ha^−1^)		Composition (%)	
		In	Un	In	Un	In	Un	In	Un	In	Un	In	Un
*Senegalia mellifera*	Fabaceae	44	321	51.2	28.9	131	250	30.3	14.9	0	0	0	0
*Vachellia hebeclada*	Fabaceae	42	273	48.8	24.6	231	121	53.5	7.2	0	0	0	0
*Vachellia erioloba*	Fabaceae	0	367	0	33.1	0	444	0	26.5	76	132	46.3	64.7
*Vachellia tortilis*	Fabaceae	0	0	0	0	0	74	0	4.4	0	0	0	0
*Ziziphus mucronata*	Rhamnaceae	0	17	0	1.5	70	103	16.2	6.1	0	0	0	0
*Grewia flava*	Malvaceae	0	124	0	11.2	0	0	0	0	0	0	0	0
*Tarchonanthus camphoratus*	Asteraceae	0	8	0	0.7	0	686	0	40.9	0	0	0	0
*p*-value		*p* < 0.05				*p* < 0.05				*p* = 0.6			

**Table 2 plants-12-01561-t002:** Results of the similarity percentage (SIMPER) analysis on species composition between *Prosopis velutina* invaded and uninvaded stands across the three sites along the Molopo River.

Woody Species	Average Dissimilarity = 78.2%				
	Invaded	Uninvaded	Av. Dissim.	Contribution (%)	Cumulative (%)
*Vachellia erioloba*	25.3	214	23.18	29.64	29.64
*Senegalia mellifera*	66	211	16.56	21.17	50.81
*Vachellia hebeclada*	51	176	13.93	17.82	68.63
*Ziziphus mucronata*	31	123	8.50	10.88	79.51
*Vachellia tortilis*	24.7	96	6.76	8.64	88.15
*Tarchonanthus camphoratus*	0	98	5.33	6.81	94.96
*Grewia flava*	0	55	3.94	5.04	100

**Table 3 plants-12-01561-t003:** Mean values/stand for ecological indices between invaded and uninvaded stands at all three sites.

Site	Invasion Categories	Number of Species (S)	Species Richness (R)	Shannon’s Index of Diversity (H’)	Simpson’s Index of Diversity (D)	Species Evenness (J)
**Tshidilamolomo**	Invaded	2	0.22 ± 0.0	0.70 ± 2.3	0.27 ± 0.1	0.70 ± 0.0
	Uninvaded	6	0.71 ± 0.8	1.41 ± 0.1	0.50 ± 0.0	1.00 ± 0.1
	% Decrease over uninvaded	67	69.0 ± 0.4	50.0 ± 1.7	31.5 ± 0.0	43.0 ± 0.0
*p*-value		*<*0.05	*<*0.05	*<*0.05	*<*0.05	*<*0.05
**Mabule**	Invaded	3	0.33 ± 0.4	1.00 ± 0.1	0.27 ± 0.1	0.75 ± 0.1
	Uninvaded	6	0.67 ± 0.1	1.50 ± 0.1	0.40 ± 0.0	0.90 ± 0.0
	% Decrease over uninvaded	50	50.7 ± 0.2	33.3 ± 0.1	17.8 ± 0.0	20.0 ± 0.0
*p*-value		*<*0.05	*<*0.05	*<*0.05	*<*0.05	*<*0.05
**Bray**	Invaded	2	0.19 ± 0.1	0.65 ± 0.0	0.54 ± 0.0	1.00 ± 0.0
	Uninvaded	2	0.20 ± 0.0	0.70 ± 0.0	0.50 ± 0.1	0.96 ± 0.0
	% Decrease over uninvaded	0	5.26 ± 0.1	7.70 ± 0.0	7.40 ± 0.1	4.20± 0.0
*p*-value		=0.05	=0.05	=0.06	=0.06	=0.06

## Data Availability

The data supporting the results of this manuscript will be made available upon acceptance for publication.

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
