# Peer review of "Impact of Prosopis velutina Wooton on the Composition and Diversity of Native Woody Species in a Semi-Arid Zone along the Molopo River, South Africa"

_plants, 2023, doi:10.3390/plants12071561_

Round 1
Reviewer 1 Report
The aims of this study was to evaluate if the invasion of P. velutina affect the composition of other native woody plant species. The topic of this article is very interesting. Moreover, the article is well written and presented useful experimental results that deserve to be published in Plants Journal after minor revision.
Main comments:
1.The authors should add a photograph about this invasive species in Material and Methods.
2.Figures 1 and 3 should be deleted since the data are also presented in Table 1 and 3.
3.The authors should explain in discussion section why Prosopsis velutina is aggressive species? More references should be added,
4. References should be added about the ecological indices that calculated in this study.
Other comments are mentioned in the attached file.

Reviewer 2 Report
The species of the genus Prosopis are important invasive species worldwide. They are present in almost all continents. For these reasons, there is a rich scientific literature on the subject.
Apart from the publications cited in the article, the reviewer would like to refer to the following additional particularly relevant literature: Stromberg et al. 1996 and Scott et al. 2014 - importance of impact on groundwater conditions, van Klinken et al. 2006 - situation in Australia, Henderson 2007 - distribution in South Africa, Pollock et al, Allelopathy, Shiferaf et al. 2004 - situation in other African countries.
Note: The cited publication by Shackleton et al: Prosopis: a global assessment of the biogeography, benefits, impacts and management of one of the world's worst woody invasive plant taxa.
Shackleton, RT; Le Maitre, DC; (...); Richardson, DM was not published in 2015 as in the bibliography, but in 2014.
A general problem in the data sets is that one site has another dominant invasive species (Melia azederach). Prosopis was defacto replaced by Melia here. These plots complicate the comparison. The question is, what influence does Melia have on the species composition in the plots and how is the invasion influenced by Prosopis. Proposal: The plots with Melia should be excluded from the analysis and replaced by plots without an additional invasive species!
Round 2
Reviewer 2 Report
I agree with all your changes! The main methodological Question was the case of Melia. The plots were deleted.